# Differential Cutaneous Thermal Sensitivity in Humans: Method of Limit vs. Method of Sensation Magnitude

**DOI:** 10.3390/ijerph182312576

**Published:** 2021-11-29

**Authors:** Yongsuk Seo, Jung-Hyun Kim

**Affiliations:** 1Sports AIX Graduate Program, Pohang University of Science and Technology, Pohang 37673, Korea; yseokss@postech.ac.kr; 2Department of Sports Medicine, Kyung Hee University, Yongin-si 17104, Korea

**Keywords:** thermoregulation, body temperature, thermal sensation, thermal comfort

## Abstract

Introduction: The method of limits (MLI) and method of level (MLE) are commonly employed for the quantitative assessment of cutaneous thermal sensitivity. Thermal sensation and thermal comfort are closely related and thermal sensations evoked from the peripheral thermoreceptors play an important role in thermoregulatory response to maintain normal body temperature. The purpose of this study was to compare the regional distribution of cutaneous warm and cold sensitivity between MLI and the method of sensation magnitude (MSM). Method: Twenty healthy men completed MLI and MSM to compare the regional distribution of cutaneous warm and cold sensitivity in the thermal neutral condition. The subjects rested on a bed in a supine position for 20 min. Next, the cutaneous thermal sensitivity of ten body sites was assessed by the means of MLI and MSM for both warmth and cold stimuli. Results: The absolute mean heat flux in MLI and thermal sensation magnitude in MSM showed significantly greater sensitivity to cold than to warm stimulation (*p* < 0.01), together with a similar pattern of regional differences across ten body sites. Both sensory modalities indicated acceptable reliability (SRD%: 6.29–8.66) and excellent reproducibility (ICC: 0.826–0.906; *p* < 0.01). However, the Z-sore distribution in MSM was much narrower than in MLI, which may limit the test sensitivity for the detection of sensory disorders and/or comparison between individuals. Conclusion: The present results showed that both MLI and MSM are effective means for evaluating regional cutaneous thermal sensitivity to innocuous warm and cold stimulations to a strong degree of reliability and reproducibility.

## 1. Introduction

Autonomic and behavioral thermoregulation are mediated by afferent inputs from central and peripheral thermoreceptors. In humans, these thermoreceptors consist of two groups: myelinated Aδ-fibers and unmyelinated C-fibers, which respond to cold and warm stimuli, respectively [1,2]. It is generally agreed that the density of thermoreceptors is not uniform, but varied across the body, with a greater overall density of cold receptors than warm receptors [1,2]. Together with the uneven distribution of thermoreceptors, previous findings also suggested that thermal inputs from different body regions are weighed differently by the central nervous system, which may explain regional differences in thermal sensation and thermal comfort under the same degree of thermal stimuli [3].

Thermal sensation and thermal comfort are known to be related closely [4] and are widely used for many bioengineering applications, including sportswear, personal protective clothing, and smart building to improve subjective comfort [5,6]. In particular, thermal sensation evoked from the peripheral thermoreceptors plays an important role in mediating thermoregulatory behavior to maintain normal body temperature and provides the first line of defense against thermal injuries and illnesses. Therefore, the assessment of thermoreceptor sensitivity is of importance in evaluating individuals’ behavioral-thermoregulation ability as well as evaluating aging- and disease-related deterioration in thermal sensory function.

For the quantitative assessment of cutaneous thermal sensitivity, two types of psychophysical techniques are commonly employed: the method of limits (MLI) and the method of level (MLE) in which the sensation threshold is determined based on subjects’ ability to react in response to given thermal stimuli. The basic principles, including the advantages and limitations of the two test methods, are well elucidated in previous research [7,8,9]. However, in addition to the fundamental limitations of psychophysical approaches, the question of which of the two methods better assesses human cutaneous thermal sensitivity is the subject of an ongoing debate. Previous studies reported drawbacks for MLI compared to MLE in terms of accuracy [8,10,11], whereas others found no significant difference in accuracy [12], or repeatability between the two methods [9,13,14].

More recent studies [15,16,17] investigating differential cutaneous thermal sensitivity across different body regions utilized a method similar to MLE in terms of applying a constant thermal stimulation. However, the method differs in that the main outcome is a sensation magnitude (MSM) using a subjective scale rather than a threshold detection. Although regional differences in cutaneous thermal sensitivity reported by this modified version of MLE are similar to those found by MLI, to the best of our knowledge, no prior study has compared the two methods.

Because of the aforementioned advantages and limitations of the two test methods (MLE and MLI), it is important to evaluate the valid and reliable measures of the most efficient (validity and reliability) testing procedure. Therefore, the purpose of this study was to compare the regional distribution of cutaneous warm and cold sensitivity between MLI and MSM. It was hypothesized that MLI and MSM would show a similar pattern of thermal sensitivity for warm and cold stimuli across the body.

## 2. Material and Methods

### 2.1. Participants

Twenty healthy men (age: 23.1 ± 2.1 years; height: 176.7 ± 4.5 cm; weight: 76.6 ± 4.5 kg) volunteered to participate in this study. The participants’ health condition was reviewed by a health questionnaire and those with a previous history of neurological or sensory disorders were excluded from the study. Before study participation, both written and oral consent were obtained from all the participants and the study protocol was approved by the Institutional Review Board (KHGIRB-19-286).

### 2.2. Experimental Procedure

The participants visited the testing laboratory on three occasions, separated by at least 48 h, for one familiarization and two experimental participations. For the study participation, they were asked to arrive at the testing laboratory at the same time of morning or afternoon, while abstaining from any strenuous activities that may affect their body temperature, such as exercise at least 12 h before a scheduled visit. The participants were also instructed to abstain from strenuous exercise, caffeine, and alcohol for at least 24 h before each experimental trial.

For the experimental trials, the participants wore athletic shorts and were instrumented with skin thermistors (ITP082-25, Nikkiso-Therm Co., Ltd. Tokyo, Japan) on ten body regions (forehead, neck, chest, abdomen, shoulder, forearm, hand, thigh, shin, and foot) where cutaneous thermal sensitivity was measured. Next, they rested on a bed in a supine position for 20 min followed by a measurement of the resting core body temperature using an infrared tympanic membrane thermometer (Welch/Allyn Pro 4000, Hill-Rom Holdings, Inc. Chicago, IL, USA). Upon the completion of the experimental preparation, the participants’ cutaneous thermal sensitivity in each region was assessed by means of MLI and MSM for either warmth or cold sensitivity. The order of the test methods and measurements across the body regions was counterbalanced, and warmth and cold sensitivity tests were undertaken on a separate day to minimize the transient effect of bidirectional thermal stimuli on the body.

The measurements of cutaneous thermal sensitivity via MLI and MSM were performed using a thermoception analyzer (Intercross-210, Intercross Co., Tokyo, Japan). The analyzer is equipped with a thermal stimulator built with a Peltier element (2.5 × 2.5 cm) for thermoelectric cooling and heating, which also measures heat flux between the stimulator and skin surface.

For the MLI, the stimulator was first stabilized to the skin temperature of each measurement site within a range of heat flux at ±30 W/m^2^. After stabilization, the stimulator temperature was either increased or decreased at 0.1 °C∙s^−1^ until the participants perceived a warm or a cold sensation, at which point they pressed a hand-held switch. The sensitivity was determined as a heat flux difference (W/m^2^ in an absolute value) between the resting state and the perception of warm or cold stimuli. Therefore, the lower the heat flux difference, the greater the cutaneous thermal sensitivity.

For the MSM, the stimulator temperature was set at either 20 or 40 °C for cold and warm stimuli, respectively. These temperature ranges were chosen based on previous findings of innocuous thermal stimuli without pain [15,16,17]. The stimulator was placed onto each skin site for 10 s while the temperature was kept constant. Next, the participants were asked to indicate the level of their temperature sensation using an 11 level scale (0: Not cold/hot; 0: Extremely cold/hot) adopted from previous studies [18].

The measurements of cutaneous thermal sensitivity by means of MLI and MSM were duplicated with a 10 min rest between the tests to minimize temperature perception bias from the previous measurement and the average values were used for the analyses. All the experimental tests were carried out in a thermoneutral environment (ambient temperature: 25 °C; relative humidity: 50%).

### 2.3. Statistical Analyses

The smallest real difference (SRD) was calculated to determine the reliability of each test method [19]. The SRD was utilized to determine the measurement error [7]. When the SRD% is less than 30%, the measurement error is acceptable [20]. Furthermore, a pairwise comparison with least significant differences (LSD) was conducted to determine the ranking of the regional sensitivity. Intra-class correlation coefficients (ICC) were used to determine the reproducibility of the repeated measurements for each test method. The ICC value was considered as follows: below 0.4 = poor reliability; between 0.40 and 0.59 = fair; between 0.60 and 0.74 = good; and between 0.75 and 1.00 = excellent reliability [21].

Finally, two-way analysis of variance (ANOVA) was performed for cutaneous warm and cold sensitivity to determine whether differences existed between the two methods. For this purpose, the MLI and MSM data were converted to Z-scores due to the unit difference between the methods. A statistical significance was set at *p* < 0.05 and all the statistical analyses were performed using the SPSS software package (v. 25.0, IBM, NY, USA).

## 3. Results

The absolute mean heat flux in the MLI was significantly lower by 167 W/m^2^ for cold sensitivity compared to warm sensitivity. Similarly, thermal sensation in the MSM was significantly higher by 1.6 points for cold sensitivity compared to warm sensitivity, indicating greater cutaneous sensitivity to cold stimuli (Table 1). The smallest real difference percentage was slightly lower in the MLI than in the MSM for both the warm and the cold sensitivity tests; however, all the values were below 10%, indicating an acceptable random measurement error, and, thereby, the acceptable reliability of each method (Table 1). The intra-class correlation coefficients for both MLI and MSM were above.80 (*p* < 0.001) on both the warm and cold sensitivity tests; therefore, the reproducibility of each method was found to be excellent (Table 1).

No significant interaction was found in warm sensitivity between the MLI and MLE (F = 2.899, *p* = 0.105), nor for the main effect of the method, although differences were found across the body regions (F = 34.267, *p* < 0.001). Similarly, no significant interaction was found in the cold sensitivity between the MLI and MLE (F = 2.757, *p* = 0.068), nor for the main effect of the method, although differences were found across the body regions (F = 37.262, *p* < 0.001) (Figure 1).

The pairwise comparison showed that the forehead was the most sensitive to both warm and cold stimuli regardless of the test method (*p* < 0.01). Further, the average Z-score was not significantly different between the MLI and MSM for both warm and cold sensitivity; however, the MSM showed a much narrower z-score distribution for cold sensitivity than the MLI (Figure 2).

## 4. Discussion

The present study showed that the absolute mean heat flux in the MLI was significantly lower during cold than warm stimulation and, similarly, a significantly greater thermal sensation magnitude was obtained with the cold stimulation in the MSM. These results are in agreement with previous findings of greater cutaneous thermal sensitivity to cold stimulation, which were attributed either singly or in combination to higher density [22], and/or to the faster rate of neural impulses of cold receptors [23,24]. Therefore, the differential sensitivity assessment of warm and cold stimulation by each sensory modality showed similar results, together with acceptable reliability and excellent reproducibility (Table 1), in the present comparison.

In the results reported in this study, regional differences in thermal sensitivity within the body were also observed, with the forehead being the most sensitive to warm/cold stimulation in both methods. This was not unexpected; it agrees with previous findings [3,25,26], probably due to differences in site-dependent thermoreceptor density [27], skin types (e.g., glabrous and non-glabrous) [28], and/or central processing [29].

Although a ranking order of regional sensitivity across the body showed a similar trend between the methods (Figure 1), the Z-score distribution for the warm/cold sensitivity tests differed between the methods in that the MSM values were more narrowly dispersed than the MLI values, especially for cold sensitivity (Figure 2). This may have been due to the fixed degree of thermal stimulation (20 and 40 °C for cold and warm, respectively) being too high to discern small differences between the regions and/or an intrinsic limitation for the assessment of thermal sensation magnitude using a categorical scale in the MSM. Further, this discrepancy may have resulted from anatomical features and properties, since these two methods rely on psychophysical perception [30]. Indeed, the forehead is located near the brain and is well vascularized with a thin layer of subcutaneous fat to maintain brain temperature [31]. A previous study reported that the forehead reached the highest mean temperature during thermoneutral, passive heating, and exercise in hot conditions [32]. Therefore, notably reduced sensation magnitude distribution may be a potential limitation when used to detect sensory disorders when comparing between healthy and diseased individuals [33].

It is worthy to note that MSM in the present study is somewhat different from the conventional MLE, which also assesses the thermal detection threshold in a reaction-time-exclusive manner similar to that of the MLI in a reaction-time-inclusive manner. The conventional MLE often requires as many as 10 performances to assess the threshold, which, in return, may provide more accurate results [7,8,10,11], while the MLI is favored for its faster and more convenient procedure than the MLE, with acceptable reliability for clinical use.

However, in the tests reported in this study, the MLI required more time to obtain an outcome value due to the skin temperature stabilization and thermal stimulation (rates of temperature change), but provided objective threshold data in the form of heat flux and temperature, which can be established and utilized as normative data for a population of interest. On the other hand, the MSM was carried out for a controlled period of 10 s for all the body regions, followed by the participants’ judgment of sensation magnitudes. Although the MSM procedure was faster than the MLI, procedural concerns were initially raised regarding temperature perception bias due to adaptive changes to constant thermal stimulation and a judgment bias for the selection among the interval scale. No direct analyses were possible to investigate these concerns; however, the fact that the pattern of regional sensitivity to warm/cold stimulation and these variances in MSM were similar to those in the MLI supports its usability for quantitative sensory testing, especially for the assessment of regional differences.

Some caution is advised for the interpretation of the present findings. First, the present study tested only young, healthy men; therefore, the results may differ from other populations when considering the previously reported effects of gender [7,15] and aging [34] on cutaneous thermal sensitivity. Second, the demonstrated data from the MSM should be interpreted concerning the specific thermal stimulation carried out in this study, which appears to be useful for determining regional differences within individuals but may not be effective at detecting functional disorders between individuals. Lastly, both methods tested in this study were carried out by stimulating a small body area via thermal conduction; therefore, stimulating a larger region of the body parts may yield different results.

## 5. Conclusions

The MLI, used in a reaction-time-inclusive manner, required a longer time to attain results, but threshold values were objectively evaluated more objectively. At the same time, the MSM, used in a reaction-time-exclusive manner using constant stimuli with a sensation magnitude scale, was more intuitive to utilize and yet provided similar results to the MLI for differential regional cutaneous sensitivity. One major limitation noted for the MSM was in the comparison between individuals by establishing normative data. Since the psychophysical approach for thermal sensory testing is to quantitatively assess one’s reaction to thermal stimulation and thereby assess a link between somatosensory afferents and subjective perception, any test may be subject to some degree of perceptual, psychological, and/or procedural bias. The results presented in this study demonstrated that both the MLI and the MSM are effective means of evaluating regional cutaneous thermal sensitivity to innocuous warm and cold stimulations to a strong degree of reliability and reproducibility.

## Figures and Tables

**Figure 1 ijerph-18-12576-f001:**
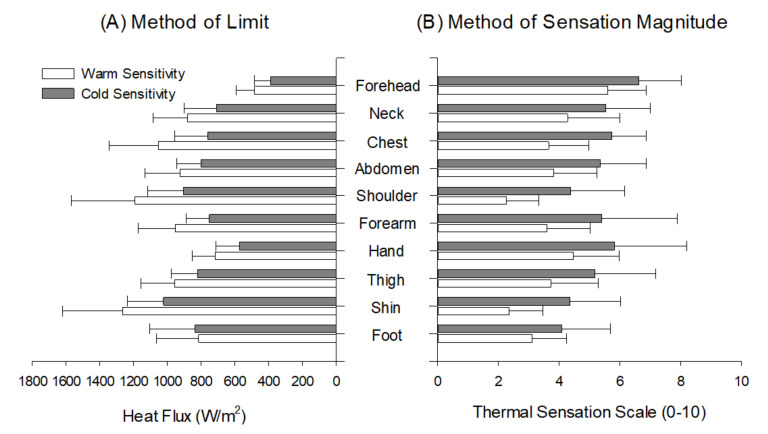
Comparison of warm and cold sensitivity of ten body regions between the method of limit and method of level.

**Figure 2 ijerph-18-12576-f002:**
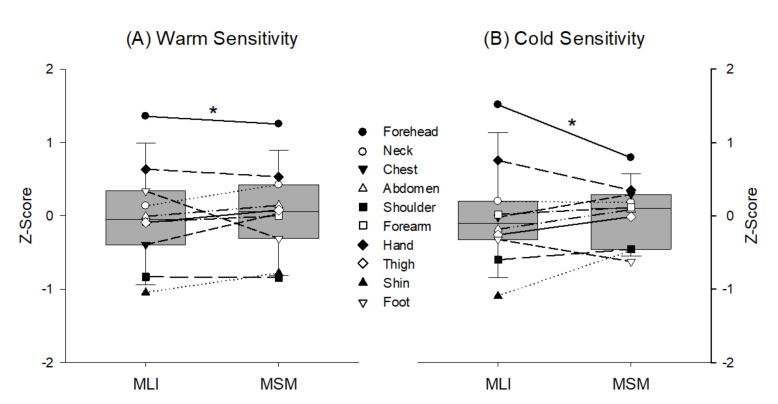
Standard score (Z-score) of warm and cold sensitivity between the method of limit and method of level across ten body regions. *: The forehead was the most sensitive to both warm and cold stimuli regardless of the test method (*p* < 0.01).

**Table 1 ijerph-18-12576-t001:** Mean absolute thermal sensitivity, reliability, and reproducibility between the method of limit and method of sensation magnitude.

	MLI		MSM	
	Warm	Cold	Warm	Cold
Mean ± SD	924.7 ± 324.3 *	757.7 ± 243.2 *	3.6 ± 1.6 *	5.2 ± 1.8 *
95% CI (upper-lower)	879.5–969.9	723.8–791.7	3.4–3.9	4.9–5.5
Standard error	22.93	17.20	0.11	0.13
Smallest real difference	63.56	47.68	0.32	0.37
SRD %	6.87	6.29	8.66	7.09
Intra-class correlation	0.826 **	0.839 **	0.906 *	0.878 **

*: Significant difference between warm and cold sensitivity tests at *p* < 0.01. **: Significant at *p* < 0.001.

## Data Availability

The data presented in this study are available on request from the corresponding author. The data are not publicly available due to the protection of personal information.

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
