# Peer review of "Differential Cutaneous Thermal Sensitivity in Humans: Method of Limit vs. Method of Sensation Magnitude"

_ijerph, 2021, doi:10.3390/ijerph182312576_

Round 1
Reviewer 1 Report
I appreciate the editors and the author the opportunity of reviewing this manuscript. In this regard, I think that the manuscript needs some work on the presentation (Introduction, Methods, Results, and Discussion - see comments attached).
The purpose of the presented study was to compare the regional distribution of cutaneous warm and cold sensitivity using the method of limits (MLI) and the method of sensation magnitude (MSM). The authors concluded that both methods are effective means for evaluating regional cutaneous thermal sensitivity to innocuous warm and cold stimulations at an appealing degree of reliability and reproducibility.
Abstract
Line 13: Why in the middle of the sentence “Method” starts with a capital letter. In line 8 the authors used small letters.
Introduction
Line 53
„…elucidated previously.” Please, include the references.
Paragraph 5
Please, provide a rationale for the usage of the MSM in thermal sensitivity assessment. What are the benefits of this method based on a subjective assessment of thermal sensation?
Material and methods
Did the participants were asked not to use any stimulants: coffee, alcohol, nicotine, other prior to examination? Such substances may influence central processing.
Line105 and 108: An incorrect unit is used. Should be W/m2 or W·m-2.
Line 120: Why is a 10-minute interval between tests adopted? Accordingly to the literature whether arbitrarily?
Line 125: Please address the scientific source for the acceptable value of SRD% as there is no commonly accepted criterion available for its judging.
Line 126: what intervals for ICC interpretation did the authors use, please include a reference.
Results
Table 1. Please, include the units. Why the MLI and Warm are in bold?
Discussion
The discussion did not clearly emphasize the importance of the MSM method in the study of the thermal sensitivity of the skin. It was shown that both methods provide similar results only. I would develop the importance, the benefits of using the MSM in cutaneous thermal sensitivity.
Author Response
We would like to thank you for the time and effort spent reviewing our manuscript. We found your comments and suggestions very helpful and strongly believe they improved the quality of this manuscript. Your original comments and our responses to those comments are listed below
Reviewer 1
I appreciate the editors and the author the opportunity of reviewing this manuscript. In this regard, I think that the manuscript needs some work on the presentation (Introduction, Methods, Results, and Discussion - see comments attached).
The purpose of the presented study was to compare the regional distribution of cutaneous warm and cold sensitivity using the method of limits (MLI) and the method of sensation magnitude (MSM). The authors concluded that both methods are effective means for evaluating regional cutaneous thermal sensitivity to innocuous warm and cold stimulations at an appealing degree of reliability and reproducibility.
Abstract
Line 13: Why in the middle of the sentence “Method” starts with a capital letter. In line 8 the authors used small letters.
Response: Thank you for pointing out this error. The capital “M” has been changed to a lower case
Introduction
Line 53
„…elucidated previously.” Please, include the references.
Response: The references have been added.
Paragraph 5
Please, provide a rationale for the usage of the MSM in thermal sensitivity assessment. What are the benefits of this method based on a subjective assessment of thermal sensation?
Response: We have added the following sentence as follows:
“Because of the aforementioned advantages and limitations of the two test methods (MLE and MLI), it is important to evaluate the valid and reliable measures of the most efficient (validity and reliability) testing procedure.”
Material and methods
Did the participants were asked not to use any stimulants: coffee, alcohol, nicotine, other prior to examination? Such substances may influence central processing.
Response: We apologize for not including these details in the original submission. We have added following sentence.
“Participants were also instructed to abstain from strenuous exercise, caffeine, and alcohol for at least 24 h before each experimental trial.”
Line105 and 108: An incorrect unit is used. Should be W/m2 or W·m-2.
Response: It has been changed to W/m2 .
Line 120: Why is a 10-minute interval between tests adopted? Accordingly to the literature whether arbitrarily?
Response: We used 10minute interval between tests to minimize an order effect from previous measurement.
Now reads as follows:
“The measurements of cutaneous thermal sensitivity by the means of MLI and MSM were duplicated with a 10 min rest between the tests to tests to minimize temperature perception bias from the previous measurement and the average values were used for analyses.”
Line 125: Please address the scientific source for the acceptable value of SRD% as there is no commonly accepted criterion available for its judging.
Response: We added the following sentence in statistical analyses.
“The SRD was utilized to determine the measurement error (Lue et al., 2017). When the SRD% is less than 30%, the measurement error is acceptable (Huang, Hsieh, Lin, & Chen, 2011).”
Line 126: what intervals for ICC interpretation did the authors use, please include a reference.
Response: We have added a reference as follow:
“The ICC value is considered as below 0.4 = poor reliability, between 0.40 and 0.59 = fair, between 0.60 and 0.74 = good, between .75 and 1.00 = excellent reliability (Cicchetti, 1994).”
Results
Table 1. Please, include the units. Why the MLI and Warm are in bold?
Response: Thank you for pointing out this error. We changed MLI and Warm in plain font.
Discussion
The discussion did not clearly emphasize the importance of the MSM method in the study of the thermal sensitivity of the skin. It was shown that both methods provide similar results only. I would develop the importance, the benefits of using the MSM in cutaneous thermal sensitivity.
Response: Thank you for the suggestion. However, we already stated the benefits of MSM based on a subjective assessment of thermal sensation in line 203-222.
Reviewer 2 Report
The Manuscript “Differential cutaneous thermal sensitivity in human: Method of limit vs. Method of sensation magnitude” demonstrated the regional distribution of cutaneous warm and cold sensitivity using the method of limits (MLI) and method of sensation magnitude (MSM) across the body. The study shows that the absolute means of heat flux in MLI and MSM have significantly higher sensitivity to cold than warm condition and both methods have a similar pattern of regional differences across the body. the study also demonstrated that both methods are reliable and reproducible measures. The authors concluded that MLI and MSM are effective tools to evaluate regional cutaneous thermal sensitivity in warm and cold conditions.
The manuscript is interesting to read and provides useful methods as assessment tools to determine the cutaneous thermal sensitivity. The manuscript is well written and has logical and progressive structure. The data is well presented. However, the research design is not clear. Overall, this manuscript merits to publish the journal with appropriate revision.
It is characterized by major and minor issues as below,
Major:
- The authors have used the two-way (RM) ANOVA. I assume that one independent variable will be method and another Independent variable will be conditions such as warm and cold. The authors also mentioned that two-way ANOVA was performed for cutaneous warm and cold sensitivity to determine if differences exist between the two methods. In this case, It might be better to use the two-way (RG) ANOVA. Please clarify why repeated-measures ANOVA was used.
- In results, the author was performed pairwise comparison (line 154 and 155) even if there are no interaction and no main effects. When authors used two way ANOVA design, pairwise comparison is not necessary.
Minor:
- In table 1 and figure 2, There are all stars on the data. Please present stars in one side when author find significance.
Author Response
We would like to thank you for the time and effort spent reviewing our manuscript. We found your comments and suggestions very helpful and strongly believe they improved the quality of this manuscript. Your original comments and our responses to those comments are listed below
Reviewer 2
The Manuscript “Differential cutaneous thermal sensitivity in human: Method of limit vs. Method of sensation magnitude” demonstrated the regional distribution of cutaneous warm and cold sensitivity using the method of limits (MLI) and method of sensation magnitude (MSM) across the body. The study shows that the absolute means of heat flux in MLI and MSM have significantly higher sensitivity to cold than warm condition and both methods have a similar pattern of regional differences across the body. the study also demonstrated that both methods are reliable and reproducible measures. The authors concluded that MLI and MSM are effective tools to evaluate regional cutaneous thermal sensitivity in warm and cold conditions.
The manuscript is interesting to read and provides useful methods as assessment tools to determine the cutaneous thermal sensitivity. The manuscript is well written and has logical and progressive structure. The data is well presented. However, the research design is not clear. Overall, this manuscript merits to publish the journal with appropriate revision.
It is characterized by major and minor issues as below,
Major:
- The authors have used the two-way (RM) ANOVA. I assume that one independent variable will be method and another Independent variable will be conditions such as warm and cold. The authors also mentioned that two-way ANOVA was performed for cutaneous warm and cold sensitivity to determine if differences exist between the two methods. In this case, It might be better to use the two-way (RG) ANOVA. Please clarify why repeated-measures ANOVA was used.
Response: We apologies for this mistake. We agree that we had two test methods (MLI and MSM) by two stimuli (warm and cold). We have change to “Two-way ANVOA”.
In results, the author was performed pairwise comparison (line 154 and 155) even if there are no interaction and no main effects. When authors used two way ANOVA design, pairwise comparison is not necessary.
Response: Thank you for pointing out this issue.We agree when a significant F ratio for main effect and interaction were not detected, post-hoc pair-wise comparison is not necessary. However, we wanted to determine the most sensitive site and rank order of regional sensitivity for both warm and cold stimuli.
We have added following sentence in the statistical analyses.
“Also, a pairwise comparison with least significant differences (LSD) was conducted to determine the ranking of the regional sensitivity.”
Minor:
- In table 1 and figure 2, There are all stars on the data. Please present stars in one side when author find significance.
Response: Thank you for pointing out these errors. The bale 1 shows a correct presentation of the statistical significance. However, Figure 2 was mistakenly presented. We have removed the stars on Figure 2.
Reviewer 3 Report
The purpose of this study was to investigate the comparison between MLI and MSM on the regional distribution of cutaneous warm and cold sensitivity. Overall, the study design seems appropriate, the manuscript is generally well driven. I have some minor comments and suggestions for the authors that I believe would strengthen the manuscript.
Line 53, 55. Please insert the references.
In the introduction section, cite the different responses of the cutaneous warm and cold sensitivity.
In the discussion section, please interpret more the huge difference in the forehead between MLI and MSM. It seems to weak interpretation.
Author Response
We would like to thank you for the time and effort spent reviewing our manuscript. We found your comments and suggestions very helpful and strongly believe they improved the quality of this manuscript. Your original comments and our responses to those comments are listed below
Reviewer 3
The purpose of this study was to investigate the comparison between MLI and MSM on the regional distribution of cutaneous warm and cold sensitivity. Overall, the study design seems appropriate, the manuscript is generally well driven. I have some minor comments and suggestions for the authors that I believe would strengthen the manuscript.
Line 53, 55. Please insert the references.
Response: We added the reference.
In the introduction section, cite the different responses of the cutaneous warm and cold sensitivity.
Response: We added the references.
In the discussion section, please interpret more the huge difference in the forehead between MLI and MSM. It seems to weak interpretation.
Response: We have added following sentence.
Further, this discrepancy may result from anatomical features and properties since these two methods rely on psychophysical perception (Wang, Kim, Normoyle, & Llano, 2015). Indeed, the forehead is located near the brain and is well vascularized with a thin layer of subcutaneous fat to maintain brain temperature (Brajkovic & Ducharme, 2006). A previous study reported that the forehead reached the highest mean temperature during thermoneutral, passive heating, and exercise in the heat (Kim, Seo, Quinn, Yorio, & Roberge, 2019).